# Yeast-Based Screening of Anti-Viral Molecules

**DOI:** 10.3390/microorganisms12030578

**Published:** 2024-03-14

**Authors:** Vartika Srivastava, Ravinder Kumar, Aijaz Ahmad

**Affiliations:** 1Department of Clinical Microbiology and Infectious Diseases, Faculty of Health Sciences, School of Pathology, University of Witwatersrand, Johannesburg 2193, South Africa; vartika.srivastava@wits.ac.za; 2Department of Pathology, University of Tennessee Health Science Center, Memphis, TN 38163, USA; 3Infection Control Unit, Charlotte Maxeke Johannesburg Academic Hospital, National Health Laboratory Service, Johannesburg 2193, South Africa

**Keywords:** screening assays, anti-viral, protease, drug discovery, virus, yeast

## Abstract

Viruses are minuscule infectious agents that reproduce exclusively within the living cells of an organism and are present in almost every ecosystem. Their continuous interaction with humans poses a significant threat to the survival and well-being of everyone. Apart from the common cold or seasonal influenza, viruses are also responsible for several important diseases such as polio, rabies, smallpox, and most recently COVID-19. Besides the loss of life and long-term health-related issues, clinical viral infections have significant economic and social impacts. Viral enzymes, especially proteases which are essential for viral multiplication, represent attractive drug targets. As a result, screening of viral protease inhibitors has gained a lot of interest in the development of anti-viral drugs. Despite the availability of anti-viral therapeutics, there is a clear need to develop novel curative agents that can be used against a given virus or group of related viruses. This review highlights the importance of yeasts as an in vivo model for screening viral enzyme inhibitors. We also discuss the advantages of yeast-based screening platforms over traditional assays. Therefore, in the present article, we discuss why yeast is emerging as a model of choice for in vivo screening of anti-viral molecules and why yeast-based screening will become more relevant in the future for screening anti-viral and other molecules of clinical importance.

## 1. Introduction

Viruses are microscopic, subcellular biological entities that mainly consist of proteins and nucleic acid [ssDNA or dsDNA or (+) ssRNA or (−) ssRNA or both DNA and RNA (for example, Leukovirus)] [1]. Some viruses, such as enveloped viruses (or eViruses), also possess lipid membranes that are derived from host cells during the process of budding or at the time of cellular exit [2]. Viruses outside the host are known as virions, are biologically inert and unable to multiply, whereas upon entering the host, these inert biological structures exploit the resources from the host cell and start multiplying. Therefore, viruses are commonly referred to as cellular or molecular parasites [3]. Almost all cell types, including human cells, are susceptible to viral infections [4]. Of more importance, viruses are considered the most abundant biological entities in our environment as they are present almost everywhere life exists [5,6,7,8]. In another study, it was estimated that out of millions of viral species, only a few thousand are characterized in significant detail [9].

Owing to their ubiquitous nature and huge diversity, we are continuously exposed to a large number and diversity of viruses. Although viruses are present externally on the skin, several viruses are reported to reside inside epithelial and other cells in the host body, for instance, in the respiratory tract, lungs, gut, and around body openings [10]. Generally, individuals with competent immune systems may better eliminate the pathogenic virus from the body compared to immunocompromised patients. Conditions such as the use of immunosuppressive drugs and chronic disease make individuals prone to clinical viral infections [11,12,13]. Apart from this, other factors that may increase the likelihood of pathogenic viral infections are the age of an individual (for example, poliovirus is mostly infectious in childhood, while smallpox virus is mostly infectious in adulthood), the surrounding population or community (vaccinated or unvaccinated), and genetic factors [14,15].

Viruses mostly have a fixed and limited range of hosts they can infect. For instance, the smallpox virus only has a human host [16], whereas rabies and coronaviruses can infect a wide range of mammals [17,18,19,20]. Moreover, some viruses such as influenza can cause pathogenic infection in both mammals and birds [21], whereas it is well known that viruses causing infection in plants do not infect animals and vice versa [22]. Similarly, the viruses of bacteria or prokaryotes do not infect eukaryotic cells [23]. Due to this limited host range and strict natural boundaries, most viruses are not pathogenic to humans and thus remain unnoticed, whereas some viruses cause a mild pathology that subsides within a few days or weeks even in the absence of significant clinical intervention. Notably, certain viruses such as ebola, hepatitis B, smallpox, dengue, poliovirus, HIV, rabies, human papillomavirus, mumps, and measles are of utmost concern [24]. In addition, the mortality rate associated with pathogenic viral infections varies widely and may range from 1–2% in the case of influenza to almost 100% in the case of rabies [25]. The clinical severity associated with pathogenic viral infections may range from mild to extreme, such as in the case of polio (caused by poliovirus), in which most people have no symptoms or mild symptoms (abortive poliomyelitis) with influenza-like and intestinal symptoms, while some individuals suffer serious symptoms, including paralysis [26,27]. Coronavirus infection can lead to lung or other organ damage, which can persist for several months to years [28,29,30].

In brief, apart from non-pathogenic viruses, there exist some viruses that cause serious threats to human health globally, and thus, all measures such as vaccination, use of anti-virals, and following safety protocols (personal hygiene, using a mask if required, social isolation or distancing) should be taken to prevent or combat these clinical viral infections [31,32,33]. Herein, our primary focus is on discussing the techniques employed for screening anti-viral compounds, and we shed light on the numerous challenges associated with these methodologies. Additionally, the emergence and importance of yeast as a reliable model for screening anti-viral molecules are also included in this article.

## 2. Why Target Viral Enzymes?

To date, vaccines remain the most common and safest way to prevent pathogenic viral infections. The complete eradication of smallpox and the almost complete eradication of polio infections show the benefits and efficacy of vaccines globally [34,35]. Even in the ongoing fight to combat coronavirus infection, vaccines remain the most reliable tool to save human lives [36,37,38]. Despite this, almost all the commercially available vaccines, including recently introduced mRNA vaccines against coronavirus infection, have several issues, including their thermolabile nature [39,40]. Different issues associated with presently available vaccines and different ways to tackle them are discussed elsewhere [39,41,42,43,44]. Although their thermolabile nature and the strict requirement of following the cold chain (the need to store and transport vaccines either frozen or at 4 °C) are of concern, frequent mutations in the surface immunogenic proteins further complicate the future of subunit vaccines. For instance, the world has witnessed frequent changes in the spike (RBD) protein of SARS-CoV-2 and the emergence of several variants of the virus during the two years of the pandemic [45,46,47,48].

Apart from this, the bio-waste generated in the form of vaccine vials, needles, and syringes is another challenge. Similarly, a mismatch of syringes/needles and vaccine vials can also be a concern, as seen during coronavirus vaccination in several countries [39]. Of importance, a single dose of vaccine/s is not always 100% effective, and several doses may be required; e.g., up to 3–4 doses of coronavirus vaccines are recommended for older people and/or front-line healthcare professionals [49]. Additionally, another significant and unavoidable concern with vaccines is the associated allergies and side effects. The ability of a given vaccine to prevent pathogenic viral infections from different variants or species of the same genus remains doubtful [50]. Furthermore, it is difficult to predict at what pace vaccine development and availability on a global scale will take place in any future pandemic. Apart from this wide gap in vaccine availability in different parts of the world, economic and political issues remain a concern affecting the success of vaccination programs [51,52,53,54]. Further to this, there are instances when vaccination can be dangerous if not given under proper recommendation. For example, the dengue vaccine is recommended only to individuals who were previously infected with the dengue virus [55,56]. In addition, in some cases such as HIV, we still do not have any effective vaccines available [57].

Hence, there is strong evidence to support the idea that relying solely on vaccines may not be sufficient, and it is imperative to explore alternative approaches for preventing or managing clinical viral infections. In light of this, the urgent requirement for the development of novel and safer anti-viral medications becomes apparent, as it will help safeguard us against future outbreaks and pandemics [58,59]. Owing to this demand, non-structural viral proteins (mostly enzymes) such as proteases remain the most suitable targets for anti-viral drug development. It is important to mention that targeting viral enzymes has its own advantages as these enzymes are relatively conserved among different variants of the same virus as well as between different species and therefore can act as important targets for clinical interventions. For example, the protease of the coronavirus is conserved in SARS-CoV-2 and MERS. Even within the different strains of SARS-CoV-2, proteases show a higher degree of conservation of key amino acid residues in the active site than non-catalytic sites in the rest of the protein [60]. The degrees of conservation of different human SARS-CoV-2 and MERS-CoV-2 proteins are shown in Table 1. The viral enzymes show a higher degree of conservation compared to other structural or surface proteins [61]. Apart from proteases, other viral enzymes such as integrase and RNA-dependent RNA polymerase can also be used as targets in drug development [62,63,64].

Notably, another reason that viral enzymes are potential targets to combat clinical viral infections is the lack of human homologs in viral genomes. This feature minimizes the risk of adverse effects and cytotoxicity in humans. In addition, the use of anti-viral compounds targeting these enzymes will offer several other advantages over vaccines such as easy application, less biomedical waste generation, and relatively easy handling [65,66,67]. Thus, the development and screening of lead molecules for their anti-viral properties will be worthwhile. It is important to mention that several viruses that infect humans use host cellular proteases or proteins for multiplication and hence small molecule protease inhibitors may not be useful in these cases; e.g., the Ebola virus uses human cathepsin B for processing its protein [68]. In such cases, vaccines remain the best option for the fight against pathogenic viral infections.

## 3. Assays for Screening Anti-Viral Molecules

With ever-increasing pathogenic viral infections and due to the lack of sufficient classes of anti-viral drugs, the identification of new anti-viral molecules bearing different modes of action, and which are safe for human use, is important from the clinical point of view. Therefore, significant efforts have been made to develop assays that allow rapid screening of anti-viral molecules. The screening is performed by employing either in vitro or in vivo assays. Although every assay is associated with some benefits and drawbacks, both types of assays are still frequently used by researchers for screening purposes. In vitro and/or in vivo assays provide possible lead candidates that are currently in different phases of clinical trials or have already entered clinical use. However, it is very important to know which screening technique is suitable for a particular type of enzyme or virus. Since the information available in the literature is limited, and the use of defined assays for specific pathogenic viruses or enzymes is not yet widely agreed upon, there is confusion amongst scientists. Therefore, from a drug discovery point of view, this article presents a comparative analysis that includes the benefits and drawbacks associated with in vitro and in vivo assays.

### 3.1. In Vitro Assay for Screening of Anti-Viral Molecules

Before delving into significant concerns regarding in vitro assays, it is essential to acknowledge the factors that contribute to the popularity of these assays among researchers. The rapid availability of high-quality genome sequences and gene annotation facilitates fast cloning, expression, and purification of viral proteins including enzymes, especially proteases, and thus allows rapid screening of anti-viral molecules using in vitro assays. Apart from this, the availability of custom peptides (acting as substrates for the viral enzymes) favors rapid bursts in the available literature of in vitro assays performed for screening anti-viral molecules. Most of the in vitro assays for screening anti-viral molecules are based on FRET (Förster or fluorescence resonance energy transfer) or fluorescence intensity signal measurement using FACS (fluorescence-activated cell sorting) or a plate reader [69,70,71]. Other features that make these types of assays attractive include the ability to use or adapt such assays to a high-throughput platform (suitable for 96- or 384-well formats), high sensitivity, the ability to control reaction conditions, and minimizing the use of animals [72]. Besides these features, sometimes these assays can be relatively economical and more rapid compared to in vivo assays. Also, in vitro assays do not suffer the problem of off-targets (or nonspecific binding or interactions), which is a common occurrence in in vivo experiments involving cells or animals/organisms.

Despite several advantages, it is important to mention that sometimes in vitro techniques do not provide an exact result. For example, the result of an in vitro assay is highly dependent on the experimental conditions, including pH, molarity, or tonicity of buffers, and any deviation from the standard operating procedure (SOP) may change the result [73,74]. Therefore, it is of the utmost importance to take extra care when performing and analyzing data from in vitro assays. The in vitro assays are usually performed by using highly purified proteins and hence fail to mimic the actual molecular or cellular environment [67,68,69]. The reproducibility of in vitro assays is a big concern [75]. Another factor to consider in such studies is the protein quantity used for analysis. Generally, the concentration of enzymes used in in vitro assays falls in the µM to nM range, which is not appropriate when compared with their actual concentrations in cells or tissues [76,77]. Whether the concentrations of other chemicals (test compound or molecules) used in the assays can be attained in the cell is another concern. During in vitro assays, various chemicals or ligands interact with the target protein, and thus it may be possible that some chemical molecules may not even enter the cell (especially if molecules are charged or polar) [78]. Furthermore, even if the chemicals enter the cells successfully, their stability within the cells may be affected as most of the in vitro assays are performed at a temperature that is different from the physiological temperature (for humans, 37 °C) [76,77]. Apart from this, the expression and purification of certain proteins for an in vitro screening process can be challenging. It is important to note that the proteins purified from bacterial systems lack posttranslational modification (phosphorylation, glycosylation, etc.), which may alter the interaction between protein and ligand and hence may significantly affect the results of an experiment [79]. On the other hand, if the proteins used in such experiments are not of high purity, the repeat (duplicate, triplicate, or quadruplicate) sample results may be contradictory. Sometimes in vitro and in vivo assays do not correspond with one another. The reason could be the limitations of in vitro experiments; for instance, researchers have observed that in the case of FRET-based assays, quenching may be caused by reaction components, or they may cause increased fluorescence of reporter molecules [80]. Owing to the limitations of in vitro assays, the use of in vivo techniques for screening anti-viral molecules is highly recommended.

### 3.2. In Vivo Assay for Screening of Anti-Viral Molecules

Generally, in vivo assays are performed in animals or cell lines that mimic the microenvironment of the natural cells and tissues and thus provide researchers with more accurate platforms for investigating biological systems [81]. In vivo assays provide valuable information on the cellular toxicities of tested molecules, and the data generated from in vivo studies can be extrapolated to the respective biological systems, thereby providing confidence in decision making. Since in vivo assays mimic the cellular or molecular microenvironment, only molecules that can enter the cells and produce the desired effect are scored. The advantages of in vivo assays are that the protein purification step is not required and sometimes the final read-through can be very simple and easy (for example, growth or changes in optical density or fluorescence measurements).

The only drawback faced with in vivo assays is the involvement of animals, which may be costly, have low throughput, and require ethical approval, which can be time-consuming [82]. However, several in vivo experiments only require the use of animal cell lines (including mammalian cell lines). Some major drawbacks of using cell lines are the high maintenance cost, the possibility of laboratory-acquired contamination, the requirement of a cell culture facility, and the requirement of sophisticated and costly instruments such as FACS. Furthermore, the generation of stable cell lines or animal models for in vivo studies may be lengthy and costly. In some instances, technical difficulties occur and assays involving whole animals cannot be adopted for high-throughput platforms, and the risk of being off-target is an important concern.

## 4. Yeast as a Screening Model

Since both in vitro and in vivo assays have several disadvantages, it is therefore important to find a system that can provide maximum benefits for both in vitro and in vivo systems. Yeast emerged as a model of choice for in vivo assays. Several features that compel the use of a yeast-based platform for screening purposes are briefly highlighted here. For example, yeasts are unicellular eukaryotic organisms that offer almost all the benefits offered by mammalian or animal cell-based assays (at least for screening purposes). Importantly, basic cell processes such as cell cycle regulation or programmed cell death and proteins are conserved or similar in almost all types of eukaryotic cells, including those of mammalian cell lines [83,84]. Therefore, yeasts are particularly suited to the study of the impact of those viral activities on related cellular activities during virus–host interactions [85]. Additionally, yeasts are easy to grow, and their maintenance is very economical compared with any eukaryotic system. In the past, yeast has been successfully used to study viruses from both basic as well as applied perspectives. For example, yeast has been used to study viral replication, structure–function analysis of viral enzymes, screening of anti-viral molecules, and development of anti-viral vaccines [86,87].

It is important to mention that, like animal or mammalian cells, yeast cells are also eukaryotic in nature. However, yeast cells differ from animal cells in several ways; for instance, yeast cells possess cell walls while animal cells do not, and generally yeast cells possess central and bigger vacuoles while their equivalent in animal cells are small but numerous lysosomes. Similar to animal cells, yeast cells also possess mitochondria and lack chloroplasts. Despite several similarities and differences, yeast offers several advantages, as mentioned above. Apart from this, yeast grows faster, with a short duplication time (90–120 min for budding yeast) compared to animal cells, which divide in 18 h or more. Yeast cells in general are smaller than animal cells [43].

The *Saccharomyces cerevisiae* yeast species is the most thoroughly studied organism or system at both the genetic and molecular levels. Since *S. cerevisiae* has been studied for years, a vast number of tools (for gene tagging, deletion, expression, and different libraries) are available for this species, which is advantageous in current and future research [88,89,90,91]. When required, metabolic pathway engineering or molecular modifications can be achieved easily. Yeast cells can be easily adapted to the needs of assays, due to their easy and responsive genetic manipulation. Thus, actual viral proteins that may be potential targets for anti-viral drug discovery can be easily expressed in this model. Yeast cells can also be used in 96-well plates and, therefore, this system is suitable for high-throughput screening processes. Similar to human cells, yeast cells also actively export toxic chemicals, but this can easily be resolved, and cells can be made more sensitive to the chemical under investigation; for instance, in the case of *S. cerevisiae*, cells can be made hypersensitive by deletion of the pleiotropic drug resistance (PDR) genes *PDR1* and *PDR3* [92]. These zinc cluster transcription factors mediate general drug resistance to many cytotoxic substances. Most yeast-based screening procedures are performed in *PDR1-* and *PDR3*-deleted strains. However, in some cases, other genes such as *PDR5* or *SNQ2* are also deleted along with *PDR1* and *PDR3* [93]. Because of all these advantages, yeasts have been successfully used over the years in several kinds of screening processes.

Technological improvements and the availability of advanced molecular tools such as CRISPR/Cas9 technology allow the manipulation of cell lines in an extraordinary way when compared with the situation a decade ago [94]. However, the complexity associated with this system, such as the need for high levels of expertise, high cost, and time consumption, gave an upper edge to the yeast-based platform for high-throughput screening. It is important to note that increasing complexity means increasing the frequency of off-target effects (due to the interaction of test molecules or compounds with non-targeted cellular proteins, these off-target effects also affect the interpretation of data and increase the chances of false positives). This, in turn, arises from the complex nature of multicellular organisms with a greater number of cellular proteins than unicellular yeast, which has fewer proteins. Unlike yeast, the tissue-specific proteome also increases this complexity, and is more common in multicellular organisms. This complexity increases from unicellular eukaryotes to multicellular eukaryotes (invertebrates) to vertebrates. Figure 1 illustrates various biological models employed to assess the effects of chemicals, in terms of whether they are beneficial or detrimental. It is worth mentioning that the level of complexity increases as we move from yeast cells to mice and other animals such as rats, rabbits, flies, and fish, and these complex animals are not suitable for high-throughput screening purposes. Another reason to use alternative models which include yeast for screening of chemicals is the strict ethics regulations of different countries and government organizations aiming to reduce the use of animals in research [95]. For example, the United Kingdom recently achieved a reduction in the use of animals in research [96]. The American Environmental Protection Agency (USA EPA) also stresses reducing the use of animals in chemical testing [97]. Several other countries and organizations have adopted these regulations too, and it is anticipated that many other government organizations and countries will also follow these regulations and investigate more feasible alternatives. The other aspect that may impact drug discovery and the use of animals in the pharmaceutical sector is the use of technologies such as machine learning and artificial intelligence; however, little is known, and this subject needs further research [98].

## 5. Screening of Viral Protease Inhibitors Using Yeast-Based Platforms

In the above sections, we discussed different attributes that make yeast an ideal in vivo model for screening diverse molecules. In this section, we discuss studies where yeast has been used for screening anti-viral molecules, both from chemical libraries as well as from natural sources including plant extracts. Interestingly, the use of yeast as an in vivo model for screening anti-viral molecules is very recent and only dates back to the early 21st century. The first study where yeast was used as an in vivo model was reported in 2003 from India, where the scientists used *S. cerevisiae* for screening anti-viral compounds [99]. The principle behind the method was to disturb the programmed ribosomal frameshift by molecules that lower viral multiplication in the yeast cells and thus rescue yeast growth in the presence of positive hits [99]. Later, in 2006, a study by Cottier and co-workers, from Switzerland, reported the use of *S. cerevisiae* for screening protease inhibitors against human cytomegalovirus (HCMV) protease [100]. The description of the assay is simple and easy to interpret as it is based on the growth inhibition of yeast in the presence of active protease. It is interesting to note that the yeast-based in vivo assay can identify the susceptibility of viral protease towards different inhibitors from different clinical isolates of HIV-1. This aids in determining whether a virus develops resistance against a given molecule or not [101]. Similarly, in another study, Benko and co-workers used fission yeast for screening protease inhibitors against HIV-1 protease. The assay is based on the rescue of yeast growth in the presence of a positive hit, and the read-through consists of increases in cell density and green fluorescent protein (GFP) intensity [102]. A list of studies where yeast was used for in vivo screening of viral enzyme inhibitors is shown in Table 2.

Recently, Alalam and co-workers used budding yeast for screening of SARS-CoV-2 protease inhibitors and the rescue of yeast growth in the presence of positive hits, and increases in cell density and fluorescence intensity were determined for the assay [104]. In another study, SARS-CoV protease inhibitors were screened by measuring the rescue of yeast growth in the presence of positive hits with protease inhibitor activity [103]. It is important to note that yeast-based assays are not only used for screening viral protease inhibitors but also for screening inhibitors of other viral enzymes. For example, a yeast-based in vivo assay was able to identify the inhibitors of coronavirus RNA cap guanine-N7-methyltransferase [105]. This suggests that apart from the screening of protease inhibitors, yeast can be used for screening the inhibitors of other important viral enzymes. Although a flip or split GFP complementation-based in vivo assay has been used in yeast to study protein localization or interaction [111,112], the use of this approach for screening anti-viral molecules using yeast cells was not mentioned. However, split GFP-based in vivo screening was successfully performed in human cell lines [74,113,114]. In Table 3, we elaborate on some of the most common pathogenic viruses which infect humans along with viral proteases which have the potential to act as targets in drug discovery.

## 6. The Bottleneck of In Vivo Assays for Viral Protease Inhibitors

Despite several studies reporting the use of whole cells (yeast or mammalian cells) for in vivo screening of viral protease inhibitors, an important associated challenge is the toxicity of expressed viral proteases. The expression of viral proteases within the cells (both yeast and mammalian cells) is highly toxic and even lethal. For example, the expression of proteases from HIV [126], poliovirus [127], hepatitis A [128], and SARS-CoV [103] has been found lethal for cells. Like eukaryotic cells, the expression of viral protease is toxic even to bacterial cells [129], and this further complicates the development and application of in vivo assays for the screening of viral protease inhibitors. Because of these complications, in vitro assays are more common for screening anti-viral molecules. Due to implications such as high cost, time, limited tools and resources, and complicated regulation of gene expression in mammalian cells, it is worth using a yeast-based system for screening purposes. In yeast, the genetic expression can be easily regulated due to responsive genetic manipulation and the availability of several promoters ranging from the strong inducible (GAL promoter, CUP promoter) to the moderate or weak constitutive (STE5 promoter). Further application of auxin degron-based protein depletion in yeast can assist in maintaining the growth of cells transformed for expressing viral protease [130,131]. Another possible approach is that the substrate of the protease can be overexpressed so that the expressed viral protease has less chance of cleaving host cell proteins, thereby minimizing the toxic effect of the viral protease. Since viral protease cleavage sequences are available, they can be utilized for determining the presence of these sequences in a yeast proteome, and also to choose species with proteomes lacking those particular sequences, or to present in the smallest number of native or endogenous proteins possible. All these reasons support the observation that yeast can overcome this bottleneck in the development of more novel in vivo assays for screening viral protease inhibitors.

## 7. Conclusions

As discussed in this review, both in vitro and in vivo assays have several drawbacks despite being proven to be valuable for the screening of anti-viral compounds. In contrast, the yeast-based screening model combines several important features of both in vivo and in vitro assays, and from a pharmaceutical point of view, proved to be a better choice for screening various molecules. Furthermore, the yeast-based systems offer a cost-effective approach to studying and screening protein targets in a direct-directed manner within a eukaryotic cellular context. It is important to mention that certain yeasts, especially *S. cerevisiae* followed by *S. pombe*, remain the most used species in viral screenings. Surprisingly, other yeasts such as *Komagataella phaffii* (formerly *Pichia pastoris*) are rarely used for screening purposes [132], even though this yeast species has already gained significant importance as a biological model [133,134,135,136]. Several molecules selected through yeast-based screening are currently in use (e.g., statin), which increases the confidence in and reliability of using this approach. Nonetheless, yeast-based screening comes with its set of limitations. Factors such as the thick cell walls of yeast may serve as barriers to compounds during a screening process. Moreover, the presence of highly expressed efflux pumps may lead to the exclusion of critical lead compounds. While both of these factors can diminish the sensitivity of a particular assay, it is clear that the approach of engineering yeast to lack efflux pumps is a viable strategy to enhance the sensitivity of any given assay [137].

Several experts have predicted that there will be frequent clinical viral pandemics soon, and therefore there is a pressing need to find better ways to approach any future pathogenic viral infections [138]. In this light, developing anti-viral therapeutic agents is a medical emergency, which demands searching for more economical, more reliable, and faster platforms to screen new natural, synthetic, or semi-synthetic molecules with anti-viral activity. The present review recommends exploring the yeast-based platform as an attractive approach for screening potential anti-viral compounds.

## Figures and Tables

**Figure 1 microorganisms-12-00578-f001:**
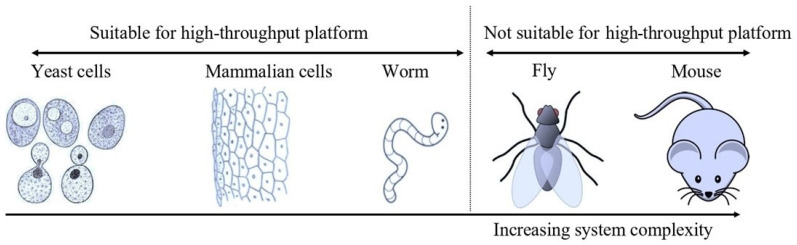
Schematic showing different models available for screening. (Note that we have not included another common model system (zebra fish) in the figure, but the overall message remains the same, namely that a complex model is not suitable for high-throughput screening). (Cartoons were taken from the mentioned site, which is properly acknowledged: http://clipart-library.com, accessed on 4 June 2022). Complexity increases from the left (yeast) to the right (mammals). So far, only yeast cells, animal cell lines, and worms are suitable for high-throughput screening (96-well format).

**Table 1 microorganisms-12-00578-t001:** Degrees of conservation between SARS-CoV-2 and MERS-CoV proteins.

Proteins	Percentage Identity
nsp1	No results after the BLAST search
nsp2	20.4
nsp3 (PL^pro^)	30.2
nsp4	40
nsp5 (M^pro^)	50.6
nsp6 (putative transmembrane domain)	34.4
nsp7 (cofactor of nsp12)	55.4
nsp8 (cofactor of nsp12)	53.0
nsp9 (RNA replicase)	52.2
nsp10	59.4
nsp11	No results after the BLAST search
nsp12 (RNA-dependent RNA polymerase)	71.3
nsp13 (helicase)	72.4
nsp14 (proofreading exoribonuclease)	62.9
nsp15 (NendoU, endoribonuclease)	50.6
nsp16 (2′-O-methyltransferase)	66.3
S (spike glycoprotein)	35.1
orf3a	No results after the BLAST search
E (envelope small membrane protein) inferred from homology	42.4
M (membrane glycoprotein) inferred from homology	42.6
orf6 inferred from homology	No results after the BLAST search
orf7a	No results after the BLAST search
orf8	No results after the BLAST search
N (nucleocapsid protein)	50.9
orf9b	No results after the BLAST search
orf10	No results after the BLAST search

**Table 2 microorganisms-12-00578-t002:** Studies where yeast was used as an in vivo model for screening of anti-viral molecules.

Yeast Species	Virus	Protease	Assay Description	Reference
*S. cerevisiae*	SARS-CoV	Papain-like protease (PLP)	Growth inhibition of yeast in (the presence of protease) rescued by inhibitor	[103]
*S. cerevisiae*	SARS-CoV-2	M^pro^	Increases in fluorescence and cell number in the presence of protease inhibitor	[104]
*S. cerevisiae*	Human cytomegalovirus	HCMV protease	Rescue of yeast by protease inhibitors by preventing cleavage of Trp1	[100]
*S. cerevisiae*	SARS-CoV	Coronavirus RNA cap guanine-N7-methyltransferase	Growth of colonies on FAO plates	[105]
*S. cerevisiae*	HIV-1	VP-1	Growth inhibition of yeast in (the presence of protease) rescued by inhibitor	[101]
*S. cerevisiae*	HIV-1	HIV-PR	Programmed—1 ribosomal frameshifting	[106]
*S. cerevisiae*	HIV-1	HIV-PR	Programmed—1 ribosomal frameshifting	[99]
*S. cerevisiae*	SARS-CoV-2	M^pro^	FACS, FRET, growth inhibition	[107]
*S. pombe*	HIV-1	HIV-PR	Rescue of growth in the presence of positive hits	[108]
*S. pombe*	HIV-1	HIV-PR	Rescue of growth in the presence of positive hits	[109]
*S. pombe*	HIV-1	HIV-PR	Rescue of growth in the presence of positive hits	[110]
*S. pombe*	HIV-1	HIV-PR	Rescue of growth in the presence of positive hits	[102]

**Table 3 microorganisms-12-00578-t003:** List of proteases from common viruses.

Virus	Disease	Nature of Genome	Enzyme Type	Protease	Reference
Polio	Polio (or poliomyelitis)	(+) ssRNA	Protease	2A^pro^ and 3 C^pro^/3CD^pro^	[115,116]
Variola	Smallpox	dsDNA	Protease	K7L	[117]
MERS	Respiratory disease	(+) ssRNA	Protease	Mpro and PLpro	[118]
SARS-CoV	Respiratory disease	(+) ssRNA	Protease	Mpro and PLpro	[119]
Dengue	Dengue	(+) ssRNA (capped)	Protease	NS2B/3	[120]
Herpes simplex virus	Cold sores, genital herpes	dsDNA (linear)	Protease	HSV protease	[121]
Varicella zoster virus	Chickenpox/varicella/shingles	dsDNA (linear)	Protease	VZV protease	[122]
Rubella	German measles or rubella	(+) ssRNA	Protease	NS-pro	[123]
Zika	Zika fever	(+) ssRNA	Protease	NS2B-NS3pro	[124]
HIV	AIDS	(+) ssRNA (linear)	Protease	HIV-PR	[125]

## Data Availability

Not applicable.

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
