# Peer review of "Yeast-Based Screening of Anti-Viral Molecules"

_microorganisms, 2024, doi:10.3390/microorganisms12030578_

Round 1
Reviewer 1 Report
Comments and Suggestions for Authors
Manuscript under review considers the yeasts as in vivo screening system for antivirals – obtained by chemical synthesis or natural (plant) products. A large review of literature is presented on the successful use of yeasts (Saccharomyces cerevisiae and some other yeasts). This approach covers especially screening of viral enzyme inhibitors.
The authors formulated the use of yeasts as a reliable in vivo system for antiviral screening thereby replacing the use of animal models for antiviral testing. The authors point out that this largely solves the problem of using animals in the selection of antiviral substances.
The presentation of the manuscript and especially its distribution into several chapters composed of rich and well-chosen research material deserves a very high evaluation.
Some remarks:
Antivirals could be classified in two groups: protein ligands and anomalous nucleosides
(modifying some of nucleic acids’ processes). This was not considered by the authors.
The authors place a very sophisticated border between screening (testing) in vitro and in vivo which is a problem in the expression of the methodology. Cell cultures system for testing is presented in several places as in vitro which is the well known, and in other places mistaken as in vivo system (p. 5, 206 and in several other places in the manuscript).
Which viruses possess both DNA and RNA? (p. 1, 31)
Smallpox infects only children (mistaken)?
What is the meaning of “the lack of human homologs in viral genoms”? (p. 3, 130)
Author Response
We would like to thank reviewer for his/her going through the manuscript and providing the positive and constructive comments. We tried to respond to reviewer comments as best we can. our point to point response to each comments are given below.
Comment: Antivirals could be classified in two groups: protein ligands and anomalous nucleosides (modifying some of nucleic acids’ processes). This was not considered by the authors.
Response: We agree with the reviewer that antivirals can be protein binding ligands as well as anomalous nucleosides which may inhibits viral genome multiplication. Infect many of anti-viral those in clinical use are anomalous nucleosides. However, we do not discuss anomalous nucleosides anti-viral in this review and left out intentionally as the main focus of review is targeting viral enzymes more specifically viral protease due to their board conservation. However, we appreciate reviewers suggestion and believes that it can be a separate manuscript.
Comment: The authors place a very sophisticated border between screening (testing) in vitro and in vivo which is a problem in the expression of the methodology. Cell cultures system for testing is presented in several places as in vitro which is the well known, and in other places mistaken as in vivo system (p. 5, 206 and in several other places in the manuscript).
Response: We are grateful to the reviewer for bring out this and we agree that this is confusing. We want to clarify that in present draft or manuscript, assays conducted in buffers outside cells or organisms are in vitro while those performed within intact cells or organism are in vivo. We also correct this in revised draft where cell-based assays are put under in vitro. Wherever cell line based were mentioned as invitro are changes to invivo in revised draft.
Note in vivo and in vitro terminology may differ in cancer or immunology field where experiments or assays performed in cell or cell lines are in vitro while those performed in animals are in vivo.
Comment: Which viruses possess both DNA and RNA? (p. 1, 31)
Response: Example of virus possessing both RNA and DNA is Leuko virus. Same example is also included in revised draft
Comment: Smallpox infects only children (mistaken)?
Response: We thank reviewers for pointing out this mistake. We modify the statement in revised manuscript. The same changes are mentioned below
(poliovirus mostly infects in childhood, while smallpox infection is more common in adults)
Comment: What is the meaning of “the lack of human homologs in viral genoms”? (p. 3, 130)
Response: Here we mean that humans lack proteins equivalent (at primary sequence, function or conserved domain or motifs) to viral enzymes. And therefore, targeting those viral enzymes will have no or very minimum toxicity or side effects.
Reviewer 2 Report
Comments and Suggestions for Authors
Srivastava and colleagues describe the emergence and importance of yeast as a reliable model for screening antiviral molecules. The review is well written, scientifically acceptable, relevant to the current scenario. However, there are several issues in the current manuscript as follows.
There is lot of discussion about the virus biology, vaccines which is not relevant to the theme of the paper.
Figure 1 does not describe the complexity of different models. Depicting an image does not explain the complexity. Complexity could have been discussed in detail.
The yeast cell model could be explained very well in terms of cell growth or other microscopic images in comparison with mammalian cell lines.
Why other viral proteins cannot used for screening is not discussed clearly.
What are the similarities and differences in terms of mammalian and yeast cells could be discussed in detail.
Author Response
We would like to thank reviewer for his/her important suggestion and constructive comments. We tried to respond to reviewer comments in as best possible ways and details as possible. Our point to point response for each comments are given below.
Comment: There is lot of discussion about the virus biology, vaccines which is not relevant to the theme of the paper.
Response: We discuss virus, virus biology and vaccines to develop a background which will help in better appreciating need for development or screening of anti-viral molecules.
Comment: Figure 1 does not describe the complexity of different models. Depicting an image does not explain the complexity. Complexity could have been discussed in detail.
Response: The purpose of figure is to inform the reader about common models available for screening purpose. Issues associated with different models are discussed in text. Information needed or relevant to models with respect to this manuscript is mentioned in text.
Comment: The yeast cell model could be explained very well in terms of cell growth or other microscopic images in comparison with mammalian cell lines.
Response: We don’t think this is needed for this review article. Comparative advantage of yeast over cell lines is discussed in text under appropriate section
Comment: Why other viral proteins cannot used for screening is not discussed clearly.
Response: Other proteins (here in present context) are structural proteins which are less conserved and are therefore less attractive as drug target. Further amino acid sequence of these proteins’ changes rapidly compared to amino acid residue at active site of protease which are highly conserved. Same is shown through table 1 by taking example of SARS-CoV-2 and MERS.
Comment: What are the similarities and differences in terms of mammalian and yeast cells could be discussed in detail.
Response: We do not discuss similarities and difference between yeast and mammalian cells as this is not the focus of the review. Similarities and difference between yeast and mammalian are discussed by others in great details several times.
Round 2
Reviewer 1 Report
Comments and Suggestions for Authors
I am satisfied with the authors responses and comments on my remarks.
Author Response
Not required
Reviewer 2 Report
Comments and Suggestions for Authors
I am not satisfied with the comments by author
Author Response
We would like to thank reviewer for his/her important suggestion and constructive comments. We tried to respond to reviewer comments in as best possible ways and details as possible. Our point to point response for each comments are given below.
Comment: There is lot of discussion about the virus biology, vaccines which is not relevant to the theme of the paper.
Response: We discuss virus, virus biology and vaccines to develop a background which will help in better appreciating need for development or screening of anti-viral molecules. This was included after suggestion from one of the reviewer
Comment: Figure 1 does not describe the complexity of different models. Depicting an image does not explain the complexity. Complexity could have been discussed in detail.
Response: The purpose of figure is to inform the reader about common models available for screening purpose. Issues associated with different models are discussed in text. Information needed or relevant to models with respect to this manuscript is mentioned in text. We further discuss the complexity in text in brief. This have been included in revised draft.
Comment: The yeast cell model could be explained very well in terms of cell growth or other microscopic images in comparison with mammalian cell lines.
Response: We briefly discuss the differences and similarities between yeast and animal cell. This have been included in revised draft. We do not go in details as these things are well discussed in general biology text books.
Comment: Why other viral proteins cannot used for screening is not discussed clearly.
Response: Other proteins (here in present context) are structural proteins which are less conserved and are therefore less attractive as drug target. Further amino acid sequence of these proteins’ changes rapidly compared to amino acid residue at active site of protease which are highly conserved. Same is shown through table 1 by taking example of SARS-CoV-2 and MERS.
Comment: What are the similarities and differences in terms of mammalian and yeast cells could be discussed in detail.
Response: In revised draft we include brief discussion on difference and similarities between yeast and animal cells.